# Urine LOX-1 and Volatilome as Promising Tools towards the Early Detection of Renal Cancer

**DOI:** 10.3390/cancers13164213

**Published:** 2021-08-21

**Authors:** Michela Murdocca, Francesco Torino, Sabina Pucci, Manuela Costantini, Rosamaria Capuano, Chiara Greggi, Chiara Polidoro, Giuseppina Somma, Valentina Pasqualetti, Yolande Ketchanji Mougang, Alexandro Catini, Giuseppe Simone, Roberto Paolesse, Augusto Orlandi, Alessandro Mauriello, Mario Roselli, Andrea Magrini, Giuseppe Novelli, Corrado Di Natale, Federica Carla Sangiuolo

**Affiliations:** 1Department of Biomedicine and Prevention, University of Rome Tor Vergata, Via Montpellier 1, 00133 Rome, Italy; miky.murdi@hotmail.it (M.M.); sabinapuc@yahoo.it (S.P.); polidoro.chia@gmail.com (C.P.); giuseppina.somma@ptvonline.it (G.S.); orlandi@uniroma2.it (A.O.); andrea.magrini@uniroma2.it (A.M.); novelli@med.uniroma2.it (G.N.); 2Department of System Medicine, University of Rome Tor Vergata, Via Montpellier 1, 00133 Rome, Italy; torino@med.uniroma2.it (F.T.); mario.roselli@uniroma2.it (M.R.); 3Department of Urology, IRCCS Regina Elena National Cancer Institute, 00144 Rome, Italy; M.Costantini@unicampus.it (M.C.); puldet@gmail.com (G.S.); 4Department of Electronic Engineering, University of Rome Tor Vergata, Via del Politecnico 1, 00133 Rome, Italy; capuano@ing.uniroma2.it (R.C.); valentina.pasqualetti@uniroma2.it (V.P.); ketchanji.mougang@ing.uniroma2.it (Y.K.M.); catini@ing.uniroma2.it (A.C.); 5Department of Clinical Sciences and Translational Medicine, University of Rome Tor Vergata, Via Montpellier 1, 00133 Rome, Italy; chiara.greggi@gmail.com; 6Department of Chemical Science and Technology, University of Rome Tor Vergata, Via della Ricerca Scientifica 1, 00133 Rome, Italy; roberto.paolesse@uniroma2.it; 7Department Experimental Medicine, University of Rome Tor Vergata, Via Montpellier 1, 00133 Rome, Italy; alessandro.mauriello@uniroma2.it; 8IRCCS Neuromed, Via Atinense 18, 86077 Pozzilli, Italy

**Keywords:** clear cell renal cell carcinoma (ccRCC), LOX-1 protein, volatile compounds (VOCs), prognostic biomarker, urine, gas chromatography mass spectrometer (GC/MS)

## Abstract

**Simple Summary:**

Renal cell carcinoma (RCC) is often late diagnosed at an advanced stage, worsening the prognosis of the patients. Thus, an early marker is desirable. This paper presents an innovative combined approach useful to identify, for the first time, the presence of LOX-1 protein within the urine of clear cell RCC patients. The LOX-1 protein is related to metabolic disorder-associated carcinogenesis and is shown to be quantitatively correlated to tumor grade and stage. The analysis of volatile compounds released by urine shows the diagnostic potentialities of volatilome and indicates that at least one volatile compound is correlated with both LOX-1 and cancer. In this work, we propose the potential use of a noninvasive approach that enables an early, routine ccRCC diagnosis and leads to a better management of the patients.

**Abstract:**

Renal cell carcinoma (RCC) represents around 3% of all cancers, within which clear cell RCC (ccRCC) are the most common type (70–75%). The RCC disease regularly progresses asymptomatically and upon presentation is recurrently metastatic, therefore, an early method of detection is necessary. The identification of one or more specific biomarkers measurable in biofluids (i.e., urine) by combined approaches could surely be appropriate for this kind of cancer, especially due to easy obtainability by noninvasive method. OLR1 is a metabolic gene that encodes for the Lectin-like oxidized low-density lipoprotein receptor-1 (LOX-1), implicated in inflammation, atherosclerosis, ROS, and metabolic disorder-associated carcinogenesis. Specifically, LOX-1 is clearly involved in tumor insurgence and progression of different human cancers. This work reports for the first time the presence of LOX-1 protein in ccRCC urine and its peculiar distribution in tumoral tissues. The urine samples headspace has also been analyzed for the presence of the volatile compounds (VOCs) by SPME-GC/MS and gas sensor array. In particular, it was found by GC/MS analysis that 2-Cyclohexen-1-one,3-methyl-6-(1-methylethyl)- correlates with LOX-1 concentration in urine. The combined approach of VOCs analysis and protein quantification could lead to promising results in terms of diagnostic and prognostic potential for ccRCC tumors.

## 1. Introduction

Renal cell carcinoma (RCC) is the seventh most common malignancy among men and the ninth among women in Europe [1,2], and its incidence has increased by about 2% per year in the last two decades. In 2018, about 100,000 new RCC cases and 40,000 RCC-related deaths were registered within the European Union [2]. There is a 1.5:1 predominance in men over women, with a peak incidence occurring between 60 and 70 years of age [3]. Risk factors for the development of RCC include smoking, obesity, poorly controlled hypertension, diet and alcohol, along with exposure to environmental agents (i.e., cadmium, benzene, trichloroethylene, and asbestos) [4]. Several hereditary RCC types also exist, with Von Hippel-Lindau (VHL) disease being the most common [4,5].

In recent years, the survival rate of patients diagnosed with RCC has improved mainly due to an increase in incidentally detected RCCs at earlier stages, better surgical techniques, and the availability of several targeted anticancer agents (mainly kinase inhibitors and immune checkpoint inhibitors) [6,7,8]. Even though several biomarkers, both in blood and urine, have been suggested as leading to the early diagnosis of RCC, none of them is reliable enough to be used in clinical practice [8]. Therefore, the evaluation of new potential molecules allowing early detection of RCC remains a challenging research area. RCC is a heterogeneous disease presenting with different histological subtypes, including clear cell (70–80%), papillary (10%), cromophobe (5%) RCC, and other rare ones (<5%). Clear cell RCC (ccRCC) is characterized by genetic mutations in factors that govern the hypoxia signaling pathway (HIF), resulting in metabolic dysregulation, heightened angiogenesis, intertumoral heterogeneity, and deleterious tumor microenvironmental (TME) crosstalk. It occurs in both a sporadic (nonfamilial) and a familial form, such as Von Hippen Lindau Syndrome (VHL), in which the HIF transcription factors upregulate the expression of several growth factors (VEGF, PDGF, and TGFα), inducing angiogenesis, proliferation, and migration, as well as the expression of numerous genes regulating glucose metabolism and oxygen transport and metabolism [9]. In fact, widespread metabolic reprogramming in glucose, lipid, and amino acid metabolism broadly contributes to the clear cell phenotype [10,11,12,13,14].

Both oxidative and inflammatory conditions constitute major risk factors for tumor development [15,16,17,18,19,20]. Lectin-like oxidized low-density lipoprotein receptor-1 (LOX-1) is one of the genes involved in pathological processes such as atherogenesis, metabolic diseases, hypertension, and tumorigenesis [21,22]. LOX-1 is a receptor protein, responsible for the recognition, binding, and internalization of ox-LDL. It represents the most relevant lipid metabolic genes consistently overexpressed in diverse cancer cell lineages, contributing to cellular transformation by activating NF-κB and the inflammatory pathway [23], and the maintenance of the transformed state at the same time. LOX-1 is primarily expressed in vascular cells and vasculature-rich organs, but its expression is upregulated by ox-LDL, inflammatory cytokines, oxidative stress, vasoconstrictive peptides, and shear stress [24]. When overexpressed, LOX-1 is able to activate HIF-1alpha and increase the expression of VEGF, MMP-2, and MMP-9, inducing the neoangiogenic and the epithelial-mesenchymal transition process [25,26,27,28]. Accordingly, both the risk factors, oxLDL, and its receptor LOX-1, have also been implicated in many aspects of cancer in several tissues, e.g., glioblastoma, osteosarcoma prostate, colon, breast, lung and pancreatic [29,30,31]. 

The role of LOX-1 is not as a directly responsible for cancer, but it supports its function in tumor progression through the combination of specific molecular pathways. Thus, due to the well-known mechanistic overlap existing in the pathobiology of atherogenesis and tumorigenesis [21,22,32], LOX-1 could represent a predisposing factor for several types of cancers.

RCC is increasingly recognized as a ‘disease of cell metabolism’, in which oncometabolites, once aberrantly accumulated, can contribute to tumorigenesis and influence tumor phenotype and progression [23,33]. It can be hypothesized that a metabolomics approach may provide the basis for a timely diagnosis of ccRCC. To explore this possibility, we investigated the volatile metabolites and searched for reliable biomarkers in urines. Urine is the straightforward choice for metabolomics studies in urinary tract diseases, and furthermore, is easily collected and stored for successive analysis. The search for biomarkers in urine has been driven by the observation that primary events in tumorigenesis are represented by the activation of lipid metabolism, together with diabetes, metabolic syndrome, and atherosclerosis.

The volatile fraction of the metabolome, the volatolome, is gaining growing interest because of the supposed simplicity of sample collection, the intrinsic non-invasiveness of measurements, and the wide availability of analytical methods. Studies evidenced that patterns of volatile organic compounds (VOCs) have been shown to be related to a vast range of phenomena observable in vitro, even at single cell level [34], and in vivo [35]. Several instrumental techniques are available for the analysis of volatolome. Gas chromatograph and mass spectrometers provide a thorough investigation of the volatolome composition. On the other hand, portable and easy to use instruments based on sensors arrays (so-called electronic noses) are also becoming available. Electronic noses have been demonstrated to be sufficiently sensitive and selective to identify diseases, analyzing various human samples such as breath [36], urine [37], and sweat [38].

The interplay between LOX-1 and volatile compounds was previously observed in cell cultures [39] and in tumor xenografts in murine models [40]. In both cases, the knockdown of LOX-1 resulted in a strong alteration of the pattern of volatile compounds. The changes are large enough to be also detected by an array of gas sensors. Moreover, previous studies showed that RCC induces an alteration of urine VOCs profile [41,42].

In this paper, we investigated the expression of LOX-1 protein and the pattern of volatile compounds in the urines of ccRCC patients. The comparison with a control group revealed the diagnostic properties of this approach, resulting on an alternative and innovative method for ccRCC diagnosis.

The LOX-1 protein shows a peculiar distribution exclusively revealed in tumor and peritumoral tissues. Moreover, both localization and quantitative expression are shown to be correlated with the tumor grade. Consequently, this scenario has a proportional relapse on the protein quantity revealed in urine samples in which LOX-1 is gradually excreted. More importantly, GC/MS results are in good agreement with previous studies of urine volatilome for RCC diagnosis [41,42]. The identification of RCC volatilome has also been carried out by a gas sensor array, revealing the presence of a specific compound correlated to LOX-1 concentration. Altogether, this approach is promising for the development of low-cost volatilome-based diagnosis of RCC.

This work reports for the first time the presence of LOX-1 protein in urine, pointing out LOX-1 as a potential and noninvasive prognostic marker in ccRCC. 

## 2. Materials and Methods

### 2.1. Patients

Forty ccRCC were collected from Regina Elena National Cancer Institute of Rome between November 2018 and August 2019, irrespectively of the clinical stage. The patients included in the study were not treated with any neo-adjuvant therapy before surgery. The mean age of the patients at the time of surgery was 62.1 (range 26–85). 43% of the patients were male, 57% were female (Table 1). All specimens used in this study underwent histological examination according to 2019 WHO Classification to confirm the diagnosis. The mean size of tumor was 5.3 cm (range 0.7–13 cm). According to the ISUP 2013 grading system, we found eight G1 cases, twenty-three G2 cases, nine G3 cases. A database was created, collecting the demographic, surgical, histological, and oncological information of each patient, including risk factors and clinical comorbidities such as diabetes, hypertension, dyslipidemia, BMI, smoking habits, and the presence of autoimmune diseases. Moreover, normal renal tissues of healthy people (*n* = 8) were collected from autoptic examination (performed within six hours from death) and resulted negative for any neoplasia and renal diseases at the macroscopic evaluation and were therefore used as control. For each patient, distant normal peritumoral tissues (NpT) were also collected and analyzed, as well as neoplasia. Histological classification was carried out on hematoxylin and eosin-stained slides. All specimens were formalin-fixed and paraffin embedded. Finally, first morning urine specimens were collected at the time of surgery from all the patients and stored at 4 °C until processing. The control group consisted of administrative employees (median age 60) working at the University of Tor Vergata, recruited on a voluntary basis by signing an informed consent during the last annual health surveillance program. First morning mid-stream clean catch urine samples have also been collected. Exclusion criteria are positive medical history of kidney diseases, hypercholesterolemia, hypertension, insulin resistance, diabetes, urinary infections, and hematuria.

### 2.2. ccRCC Cell Line and Immunocytochemistry (ICC)

Human ccRCC cell line 786-O (ATCC: CRL-1932) was used and grown in Dulbecco’s modified Eagle Medium (DMEM, GE healthcare, Milan, Italy) supplemented with 10% fetal bovine serum (FBS) (Euroclone, Milan, Italy), Glutamine (Euroclone, Milan, Italy), non-essential Amino Acids (Gibco, Life Technologies Corporation, Carlsbad, CA, USA), Penicillin-Streptomycin (Gibco, Life Technologies Corporation, Carlsbad, CA, USA).

The cells were plated in 4 well/chamber-slides at a concentration of 10,000 cells/cm^2^, 3500 cells/cm^2^ and 10,000 cells/cm^2^ respectively. After an overnight culture, the medium was removed and cells were fixed in formalin 10% for 5 min and stored at 4 °C in PBS1X, to perform ICC analysis. The cells were permeabilized with 0.5% Triton X-100 and 0.05% Tween-20 in PBS1X. Afterwards, cells were washed two times with PBS1X, and non-specific sites were blocked with serum incubation (ScyTek Laboratories, Super Block, Biotech Life Science reagents, Logan, UT, USA) for 5 min. Without washing, cells were incubated with primary antibodies, diluted in TBS1X/BSA 2%, for LOX-1 (Abcam, Cambridge, UK) and VEGF-A (Santa Cruz Biotechnology, Dallas, TX, USA) detection. To assess the background staining, a negative control was carried out without addition of primary antibody. After 1 h and three Logan, UT, USA) for 15 min. Finally, after the additional three washes, the cells were incubated with 3-Amino-9-Ethylcarbazole (AEC) for signal detection, and counterstained with hematoxylin.

### 2.3. Immunohistochemistry (IHC)

Serial 5 μm thick sections from formalin-fixed and paraffin-embedded specimens were deparaffinized and rehydrated through xylene and alcohol. Endogenous peroxidases were blocked in methanol solution with 3% hydrogen peroxide for 20 min; after this, sections were placed in water for 5 min. Subsequently, sections were placed for three times in washing solution TBS1X/0.1% Tween-20 (5 min each time). Then, sections were incubated for 5 min at room temperature with serum (ScyTek Laboratories, Super Block) for blocking non-specific sites, and later with LOX-1 (Abcam, ab60178) and VEGF-A (Santa Cruz, sc-7269) primary antibodies, diluted in TBS1X/BSA 2%. After one hour, sections were washed with TBS1X/0.1% Tween 20 solution as above, and then incubated for 15 min at room temperature with secondary antibody (ScyTek Laboratories, UltraTek Anti-Polyvalent Biotinylated Antibody). After repeating three washes, sections were incubated with Streptaividine solution (ScyTek Laboratories, UltraTek HRP) for 15 min. The staining was completed after a short incubation with a freshly prepared substrate-chromogen, 3-Amino-9-Ethylcarbazole (AEC) (ScyTek Laboratories). The sections were washed extensively in water and nuclei were counterstained with hematoxylin. The intensity of LOX-1 and VEGF-A staining was scored as negative/weak (−), moderate (+), and strong (++). 

### 2.4. Quantitative Measurement of Urine LOX-1 Levels

Fresh first morning urine samples (50 mL) were collected in sterile containers and processed within 1 h after collection. Urinary cells and debris were removed by centrifugation at 2000× *g* for 30 min at 4 °C, obtaining the first fraction cell-free urine. Then, only 50 mL of the collected supernatant were transferred to clean tubes and centrifugated at 10,000× *g* for 30 min at 4 °C to eliminate large microvesicles. After this centrifugation, the urine supernatant was added to a Vivaspin centrifugal concentrator (Sartorius) and then centrifugated at 3000× *g* for 30 min, in order to concentrate urine proteins.

Human LOX-1 levels were assessed from human urine RCC (*n* = 30) and ctr (*n* = 25) with Human Lectin like Oxidized Low Density Lipoprotein Receptor1 (LOX-1) ELISA kit (MB52703808, Mybiosource, Inc. San Diego, CA, USA), in accordance with the manufacturer’s instructions. The absorbance was measured with the spectrophotometer Multimode detector DTX 880 (Beckman Coulter, Milan, Italy). 

### 2.5. Western Blot of LOX-1 in Urine

An equal proportion of the original urine volume was loaded in 2X Laemmli buffer with 40 mM of dithiotheritol (DTT) on 10% SDS gels. Protein was transferred to a PVDF membrane (Hybond P, Amersham GE Healthcare, Chalfont St. Giles, UK). Anti-LOX-1 (ab60178, Abcam, Cambridge, UK) mouse monoclonal was used as primary antibody. Peroxidase conjugated secondary antibodies were used. Signal was scanned and quantified on Image Quant Las 4000 System.

### 2.6. Gas Chromatography Mass Spectrometry (GC/MS)

The supernatant fraction of urine samples, obtained as described above, was stored at −20 °C until GC-MS. The day before measurement, samples were transferred at 4 °C and maintained at this temperature overnight, in order to allow for slow defrosting. Four mL of urine supernatant were picked up, transferred in 20 mL Headspace-glass vial (SUPELCO, Bellefonte, PA, USA) and sealed using an aluminum crimp cap with PTFE/silica septum (SUPELCO, Bellefonte, PA, USA). Samples, thus prepared, were kept at 4 °C until VOC analysis. For each subject, two urine vials have been prepared: one for GC/MS and one for gas sensor array analyses. Urine samples were taken out of the fridge and left at room temperature for 5 min before GC/MS analysis. Then, vials were thermostated in water bath at 50 °C for 10 min in order to reach VOC equilibrium in urine headspace, using the adjustable heater C-MAG HS 7 IKAMAG coupled to ETS-D5 thermometer (IKA^®^-Werke GmbH & CO. KG, Staufen, Germany). The adsorption of urine volatile compounds was carried out by Solid Phase Microextraction (SPME) technique. A 50/30 μm divinylbenzene/carboxen/polydimethylsiloxane (DVB/Carboxen^®^/PDMS- SUPELCO, Bellefonte, PA, USA) fiber was manually exposed to sample headspace for 1 h at 50 °C. Sampled volatile compounds thermally desorbed from the fiber at 250 °C for 3 min in the GC injection port. Chemical analyses were performed by means of GCMS-QP 2010 Shimadzu series gas chromatograph mass spectrometer (Shimadzu, Kyoto, Japan), equipped with EQUITY-5 (poly(5% diphenyl/95% dimethyl siloxane) phase, SUPELCO, Bellefonte, PA, USA) capillary column, 30 m length × 0.25 mm I.D.× 0.25 μm thickness. Analysis was performed in splitless mode using ultra-high purity helium as carrier gas. VOCs were separated on the GC column using an initial oven temperature of 40 °C for 5 min, then increased by 5 °C/min to 220 °C, held for 2 min, ramped to 250 °C at 7 °C/min and lastly at 20 °C/min to 300 °C and held for 3 min (total run time: about 48 min). 

The instrument was controlled in linear velocity constant mode, using a carrier gas pressure of 24.9 kPa, total flow of 5.9 mL/min, column flow of 0.7 mL/min and linear velocity of 30.2 cm/s. 

The mass spectrometer was a single quadrupole analyzer operating in electron ionization mode, with an ionization energy of 70 eV, scanning over a mass range between 30 and 400 *m*/*z* in full scan mode. The temperature of interface and ion source was kept constant at 250 °C.

The GC-MS data have been analyzed using the section GCMS post-run analysis of the GCMS solutions software (version 2.4, Shimadzu Corporation, Kyoto, Japan). Putative identification of compound was carried out by a comparison of mass spectra with both NIST 127 and NIST 147 libraries.

In order to avoid any contamination from previous uses, SPME fibers were conditioned before exposition to the first urine sample of the day, according to supplier guidelines.

### 2.7. Gas Sensor Array Analysis

The urine samples have been maintained room temperature for 10 min before analysis, in order to achieve thermodynamic equilibrium in the headspace. Two needles were inserted into urine headspace through vial cap septum. One needle was connected to the gas sensor array via a 30 cm tube PUN-03 (FESTO, IT), and used to uptake headspace portion, and the other to a calcium chloride dryer to avoid the formation of a vacuum in the vial during the measurements.

The instrument used for this study is the latest version of gas sensor array developed at the University of Rome Tor Vergata. The array consists in an ensemble of twelve quartz microbalances (QMBs) with a fundamental frequency of 20 MHz and functionalized using porphyrins and corroles. In low-perturbation regime, a mass change (Δm) in a sensing layer coating quartz surface results in a proportional frequency variation (Δf). The sensor configuration used in this work was adopted in a previous study on LOX-1 role in colorectal cancer prognosis [40]. 

Sensor baseline signals were obtained using a constant flow of 50 sccm (standard cubic centimeter per minute) of environmental air filtered by a CaCl2 trap. The sample headspace was analyzed by the sensor array for 60 s. The difference between the oscillation frequency of the headspace sample and the reference air is the considered response of each QBM sensor, expressed in Hz.

The sensor system control and data acquisition were managed by an in-house software running in Matlab.

### 2.8. Statistical Analysis

For Western blot, IHC and ELISA assays statistical analyses were performed by the SPSS program, version 25 (IBM Corp, Armonk, NY, USA). All values provided in the text and figures are means of three independent experiments ± standard deviations (SD).

The statistical differences of GC/MS peaks abundances and sensors data between tumor and non-tumor urines samples were evaluated with the Kruskal–Wallis rank sum test.

The absolute abundances of GC-MS identified peaks and the sensor responses were arranged in matrices and analyzed with multivariate data analysis. Principal component analysis (PCA) and Linear Discriminant Analysis (LDA) were calculated on standardized data matrices where each variable of the matrices (GC/MS peaks and sensors signals) were normalized to null mean and unitary variance. 

LDA classifiers were optimized with a k-fold cross-validation procedure. Accuracy, sensitivity, specificity, and the area under the curve were evaluated to assess the performance of the classification. To avoid overfit the LDA classifier was tested with a dataset not used in training. For this scope, the sets of GC/MS and sensor array data were randomly split in training and test sets with 30% of samples used for tests. To avoid any bias in the split of data in the two groups, LDA models were calculated 100 times on random training and test partitions, and the average results were considered as the representative performance of the classifiers.

All calculations were performed in Matlab R2021a, PCA and LDA were calculated with the Statistics and Machine Learning toolbox of Matlab.

## 3. Results

### 3.1. LOX-1 Is Expressed in a Representative ccRCC Cell Line

Regarding the preliminary test, LOX-1 expression was detected in 786-O by immunocytochemistry, as shown in Figure 1, evidencing a prevalent nuclear localization (++). In parallel, a similar expression pattern was observed for VEFG-A, modulated by LOX-1, according to literature data [40]. VEGF-A is expressed in a dotted fashion, mainly related to the nucleus (+/++), as compared to the cytoplasm (+).

### 3.2. LOX-1 Is Overexpressed in the Extracellular Space of G2 and G3 Tumors

Tumoral tissue examination revealed kidney parenchyma with a focus on infiltrative clear cell population comprised of bland looking single cells or small cohesive nests, surrounded by rich capillary network. Tumoral cells contained slightly eccentric nuclei with inconspicuous nucleoli visible only at high magnification, and a cytoplasmic glycogen storage.

In order to evaluate LOX-1 expression in clear cell renal carcinoma at different grades and stages, immunohistochemistry was performed on 40 human clear cell renal cancer tissues; healthy tissues aside the neoplasia (called Normal peritumoral Tissue; NpT) from the same patients were also examined. Additionally, eight autoptic healthy kidney tissues were examined for LOX-1 expression, as described in Section 2. The histological features of the autoptic healthy kidney tissue are shown in Figure 2A–C, in which hematoxylin/eosin staining allows to appreciate the renal tubules at two different magnifications. An immunohistochemical investigation showed that LOX-1 expression was completely absent in all autoptic healthy tissues analyzed (*n* = 8) (Figure 2B–D).

In contrast, differential LOX-1 expression was found in NpT tissues according to tumor grade. In NpT G1 tissues (*n* = 8), a clear expression of LOX-1 was observed exclusively in the cytoplasm (++) (NpT G1-nucleus vs NpT G1-cytoplasm: *p* < 0.0001) (Figure 3A). Conversely, LOX-1 expression was detected both in the nucleus (+) and weakly in the cytoplasm (+/−) in the tumor counterpart (Tumor G1-nucleus vs Tumor G1-cytoplasm: *p* < 0.001) (Figure 3B, Table 2). In addition, a discrete expression of LOX-1 can also be observed in the extracellular space (++) (Tumor G1-nucleus vs Tumor G1-extracellular space: *p* < 0.01) (Table 2, Figure 3B). The pattern of LOX-1 expression found in NpT G1 tissues remained the same in NpT G2 (*n* = 23) (Figure 3C), as LOX-1 expression is related only to the cytoplasm, although less evident (+) (NpT G1-cytoplasm vs NpT G2-cytoplasm: *p* < 0.001). This expression was different in the tumor counterpart: 12 of 23 G2 cases showed only 10% of nuclei weakly positive for LOX-1 expression (+/−), while the signal was completely absent in the cytoplasm. In contrast, strong LOX-1 expression is observed in the extracellular space (++) (Tumor G2-nucleus vs Tumor G2-extracellular space: *p* < 0.0001; Tumor G2-cytoplasm vs Tumor G2-extracellular space: *p* < 0.0001). (Figure 3D). Finally, in NpT aside G3 tissue (*n* = 9), LOX-1 expression was detected again exclusively in the cytoplasm (NpT G3-nucleus vs NpT G3-cytoplasm: *p* < 0.001) (+) (Figure 3E), while the tumor counterpart lost the weak nuclear positivity found in G2 and consequently showed a strong positivity in the stroma (++), suggesting that this protein may acquire an extracellular localization, being released from tumor cells (Tumor G3-nucleus vs Tumor G3-extracellular space: *p* < 0.0001; Tumor G3-cytoplasm vs Tumor G3-extracellular space: *p* < 0.0001) (Figure 3F) (Table 2).

The VEGF expression was barely detectable in NpT aside G2 tumor (Figure 4A), while it was significantly detected in G2 tumor tissue, as shown in Figure 4B. LOX-1 expression correlates with VEGF-A in the corresponding tissue (Figure 4C: NpT; Figure 4D: tumor), suggesting a correlation between the two proteins.

For validation purposes, an analysis of the TCGA ccRCC dataset was performed reporting a progressive increase of *OLR1* gene expression in tumoral and peritumoral tissue with respect to healthy tissue. The analysis is available at the following link (accessed on 20 July 2021): https://www.ebi.ac.uk/gxa/experiments/E-MTAB-5200/Results?specific=true&geneQuery=%255B%257B%2522value%2522%253A%2522OLR1%2522%252C%2522category%2522%253A%2522symbol%2522%257D%255D&filterFactors=%257B%2522DISEASE%2522%253A%255B%2522renal%2520cell%2520carcinoma%2522%252C%2522normal%2520-%2520adjacent%2520to%2520renal%2520cell%2520carcinoma%2522%252C%2522normal%2520-%2520cortex%2520of%2520kidney%2520%28GTEx%29%2522%255D%257D&cutoff=%257B%2522value%2522%253A0.5%257D&unit=%2522TPM%2522. 

### 3.3. LOX-1 Levels Are Impaired in Clear Cell Renal Cancer (ccRC) Urine Patients

With the aim of verifying whether LOX-1, which seems to be progressively accumulated in the extracellular area according to tumor grade, was finally released, we analyzed and quantified the protein in urine collected from ccRCC patients and healthy controls by Western blot and enzyme-linked immunosorbent assay (ELISA).

Firstly, we performed a Western blot to evaluate a possible presence and accumulation of LOX-1 protein in urine. The analysis has suggested that the expression of ox-LDL receptor protein was higher in urine of patients compared to control group (Figure 5A and Appendix A), as revealed by densitometric analysis on the mean value of tumors versus healthy controls (Figure 5B; *p*-value = 0.05). 

In order to more precisely confirm this uncovering, we quantified the levels of LOX-1 in urine by ELISA, following procedures described in M&M. This analysis revealed a marked increase of LOX-1 levels in patients’ urine (*n* = 30) compared to control group (*n* = 25) (Figure 5C). Although a discrete heterogeneity was present within groups, also in this case the data results to be statistically significant (*p*-value < 0.05). Moreover, the increasing trend of protein quantity results to be proportionally correlated to the tumoral grading.

### 3.4. Volatile Compounds Analysis in Urine

The volatile compounds released by urines were analyzed with the purpose of determining a set of volatile compounds whose pattern can selectively discriminate kidney cancer samples from wild type ones.

Urine samples from 40 subjects were collected and analyzed with gas chromatography mass spectrometer (GC/MS) and a gas sensor array in order to determine a set of volatile compounds whose pattern can selectively discriminate kidney cancer. In the samples, GC/MS detected 98 volatile compounds. The majority of these compounds appeared only in some of the samples. Thus, in order to search for VOCs as univocally representative of the difference between cancer and control urines, the analysis was limited to those compounds present in at least 70% of samples. These eight compounds, listed in Table 3, were putatively identified by library comparison of mass spectra. The distribution of the abundance of each of these compounds in the categories of ccRCC kidney cancer and control is shown in Figure 6. Kruskal–Wallis rank test was used to estimate the null-probability (*p*-value) of the difference between the two categories. Table 3 shows the *p*-value of each compound. All these compounds had been previously found in human samples. The most discriminant compounds (4-heptanone, cyclohexanone, 1-hexanol-2-ethyl, and phenol) were previously indicated as potential biomarkers of kidney cancer [41,42]. In particular, cyclohexanone and 1-hexanol-2-ethyl are more abundant in the urine of cancer patients. Beside the relationship with kidney cancer, 3-methyl-butan-2-one and 4-heptanal are typical components of breath and urine [43], and the abundance of 2-nonanone in breath was found to be correlated with lung cancer [44]. 

Finally, 2-Cyclohexen-1-one, 3-methyl-6-(1-methylethyl)- has been indicated as a product of metabolism of ketones. This compound was found in rats as the main metabolic product of (R)-5-Methyl-2-(1-methylethylidine) cyclohexanone, a component of mint essential oil widely used as flavoring agent [45]. 

As shown in Figure 7, the logarithm of the abundance of 2-Cyclohexen-1-one, 3-methyl-6-(1-methylethyl) is linearly correlated with the concentration of LOX-1 in urine. Noteworthy, this is the only one of the eight selected compound showing correlation with LOX-l levels.

GC/MS analysis was complemented by a gas sensor array. The Kruskal–Wallis rank test was applied to evaluate the capability of each sensor to discriminate between cancer and wild type urine. Figure 8 shows the box-plots of sensors responses. The title of each plot shows the corresponding *p*-value. Four sensors (labeled as 3, 4, 10, and 12) are characterized by *p* < 0.05.

A better appraisal of the diagnostic properties of volatile compounds as detected by GC/MS and sensor arrays are achieved by classification algorithms. Linear Discriminant Analysis (LDA) was applied to GC/MS and sensors array data. The datasets have been split in two parts, one used to train the model and the other to test. A random split of data in two groups may result in favorable conditions that could lead to optimistic conclusions. To avoid this drawback, LDA has been calculated 100 times, each time with a different partition of data in training and test. Accuracy, namely the percentage of correct classification of test data, has been considered as the indicator of the goodness of the model. Eventually, the LDA model corresponding to the average accuracy was retained as representative for the classification.

Figure 9 shows the canonical variables and the area under the ROC of both GC/MS and sensor array classification models while the results of classification are listed in Table 4.

## 4. Discussion

RCC represents around 3% of all cancers, with the highest incidence occurring in Western countries [1,2]. During the last two decades until recently, there has been an annual increase of about 2% in incidence both worldwide and in Europe [1,2]. Verified risk factors for RCC include age, smoking, obesity, poorly controlled hypertension, diet, alcohol, and environmental and genetic factors (i.e., VHL mutations) [4]. Unfortunately, the majority of RCC (over 63%) are diagnosed when the disease is locally advanced or at a metastatic stage, and approximately two third of cases shows unfavorable prognostic factors (i.e., ISUP grade) at the diagnosis. In Europe, the 5-year overall survival rate for all types of RCC is 49% [7,8], which has recently showed important improvements mainly deriving from an increase in incidentally detected RCCs at earlier stages, better surgical techniques, and the availability of several targeted anticancer agents. Notably, diagnostic tools allowing an early diagnosis of RCC are unavailable in current clinical practice. Therefore, the search for reliable biomarkers for the early detection of RCC remains a major challenge.

Although almost all studies on RCC tissue biomarkers have been highly promising, almost all of these are based on retrospective series with small sample size and relatively short follow-up [46]. The reliability of the assay used for marker detection represents another limitation of these studies [46]. Interestingly, the identification of molecular markers in body fluids (e.g., sera and urine), which can be used for screening, diagnosis, follow-up, and monitoring of drug-based therapy in RCC patients is one of the most ambitious challenges in oncologic research [46,47]. The global analysis of gene and protein expression profiles of biological specimens, like tissue, blood, or urine, is an emerging promising tool for new biomarker identification. The markers identified from these high throughput integrated “omics” technologies have promising potential but application in clinical dia- gnostics and practical improvement in disease management is not routine [48,49,50]. Blood and urine are an ideal source of biomarker for theoretical, methodological, and practical reasons [51]. There have been some promising reports about potential biomarkers in sera, but the available data are insufficient to justify their routine clinical application [8]. Moreover, there is limited literature with regard to urine markers for RCC [8,48]. Despite this, compared with most solid malignancies, urine is the body fluid in which a RCC biomarker can be more easily and directly excreted and detected to obtain an earlier diagnosis [48]. A rational approach toward the choice of reliable biomarker(s) aimed at an early diagnosis of malignancies is focusing the research on molecules that link various risk factors and have a key role in the carcinogenesis of the disease. 

*OLR1* (OMIM#602601) is a metabolic gene that encodes for the Lectin-like oxidized low-density lipoprotein receptor-1 (LOX-1), a chameleon major receptor with a key role in the development of hypertension, diabetes mellitus, hyperlipidemia, obesity and its complications [21,31]. Several studies shed light on its role also in the stimulation of the expression of proangiogenic proteins, including NF-kB and VEGF [52]. 

To date, there is now no doubt on the function exerted by LOX-1 in tumor insurgence and progression and mostly in related neo-angiogenesis of different human cancers, such as glioblastoma, osteosarcoma, prostate, colon, breast, lung, and pancreatic carcinoma [31]. The ox-LDL receptor is a fine-tuned interplay between lipid metabolic regulator, angiogenesis, and EMT-inducing transcription factor, involved in the regulation of cancer metastasis. This suggests for the transmembrane receptor an interesting linking role between atherosclerosis, its various metabolic pathways and tumorigenesis [31]. 

In this study, we evaluate and correlate the presence and the levels of LOX-1 protein in ccRCC patients at different tumor stage in order to identify a marker for early kidney cancer detection by non-invasive methods. 

In vitro analysis of ccRCC cell lines showed that LOX-1 and VEGF are expressed both in the cytoplasm and in the nucleus. This finding prompted us to analyze 40 human kidney tumors and 8 autoptic healthy kidney tissues. 

LOX-1 was completely absent in normal autoptic renal tissues, while a staining was found only in the cytoplasm of normal peritumoral tissues (NpT) aside tumors of all grade.

These data indicate that in normal kidney tissues its function is not requested. The presence of LOX-1 in NpT can be explained considering that the NpT is perfused by soluble mediators released by the stress conditions in uncontrolled growth tumoral masses that are known to be related to LOX-1 expression [53].

With regard to the tumor counterpart, in ccRCC at different grade and stage, a differential expression of LOX-1 correlated with tumor grade was found. The expression was both nuclear and weakly cytoplasmatic in G1, while G2 and G3 showed no expression in the nucleus or the cytoplasm. Interestingly, in all three grades, LOX-1 appears to be gradually released into the extracellular space to localize in the stroma, much more evidently in G2 and G3. The level of extracellular expression also seems to correlate with tumor grade. 

Considering these data, we move to assess the presence of LOX-1 protein within urine, a peculiar circumstance never described until now. Quantitative analysis performed by ELISA demonstrated higher LOX-1 levels in ccRCC urine than in control ones in a statistically significant manner. Protein quantity is directly proportional to tumor grade. We hypothesized that one of the most likely carriers could be exosome, already involved in cancer signaling. Exosomes are a subset of tiny extracellular vesicles manufactured by all cells and present in all body fluids. During cell growth, exosomes are actively produced and released, in order to promote tumor growth, progression, and metastatic spread [54]. Moreover, it has been demonstrated that membrane cholesterol-lowering drugs induce the membrane removal of LOX-1 surface receptor within exosome membranes [55]. This hypothesis needs further experiments to be sustained. 

Cells that make up a tumor are highly metabolically active and these compounds arising from the tumor can easily cross from the cells into the urinary space, making this fluid ideal for the metabolomic discovery of RCC biomarkers. Urine is an abundant biofluid that can be readily obtained by noninvasive means. Both aspects make urine a practical choice for developing a method of early diagnosis for renal cell carcinoma [56]. 

The analysis of volatile compounds is in agreement with previous findings. Although different instruments and methods were used, some of the biomarkers found by GC/MS data corresponds with those found in previous research [41,42].

GC/MS analysis have been carried out by collecting VOCs onto SPME fiber. This is a solid phase where volatile compounds are accumulated in order to be released in the GC/MS. SPMEs are available in different materials which are oriented to maximize the accumulation of molecules with specified features. In this study, a general purpose SPME has been used. This is an optimal choice for untargeted metabolomics; however, we cannot exclude that other SPMEs could provide additional information about the relationship between volatile compounds and kidney cancer. 

Only one of the selected compounds has been found to be correlated with the concentration of LOX-1. Interestingly, this compound has been indicated as a product of the metabolism of ketones [45] and thus confirming its association with LOX-1 in turn linked to metabolism. 

On the contrary, the lack of correlation of the other volatile biomarkers with LOX-1 indicate the existence of additional pathways for these compounds. Importantly, it also indicates that LOX-1 and VOCs provide independent information that can be used to improve the diagnosis of kidney cancer.

The same samples analyzed by the GC/Ms have also been measured with a gas sensor array. These instruments, also known as electronic noses, have been frequently used to classify samples characterized by a complex chemical composition such as biological samples. The results of electronic noses are limited to the classification, so they do not provide information neither about the composition nor about the abundance of compounds. Nonetheless, for their simplicity and relative low-cost they have been indicated for routine analysis. The results show that in terms of identification of kidney cancer in respect to controls, GC/MS and electronic nose gave comparable results. So, it may be concluded that electronic noses could be used as a diagnostic tool for kidney cancer. Of course, these are the results obtained with this particular dataset. 

A manifold of sources contributes to urines composition: food, lifestyle, and drug uptake are expected to determine the chemistry of urine. Thus, environmental influences, repeatability, and long-time performance of the instrumentation need to be verified in multicentric studies. Furthermore, studies with larger sample sizes are necessary to elucidate if and how co-morbidities influence the urine volatilome.

## 5. Conclusions

Dosing LOX-1 in urine represents a novel, cost-effective, sensitive, fast, and reliable strategy for RCC diagnosis based on an ultra-sensitive and ultra-selective noninvasive tool that, in combination with other tools such as VOCs, could transform clinical management by enabling early detection of RCC and reducing unnecessary kidney biopsies and nephrectomies [57].

In the end, given the complexity of cancer treatment, it will likely require a combination of clinical and biologic approaches to fully realize the potential of precision oncology.

## Figures and Tables

**Figure 1 cancers-13-04213-f001:**
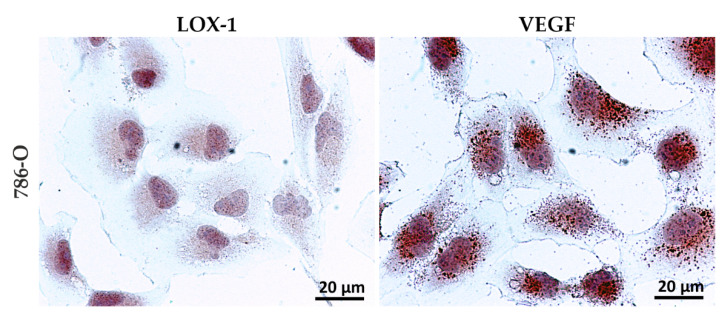
Immunocytochemistry analyses in 786-O cells. LOX-1 (**left**) and VEGF-A (**right**) immunocytochemistry analyses in 786-O ccRCC cell line. Scale bar 20 µm.

**Figure 2 cancers-13-04213-f002:**
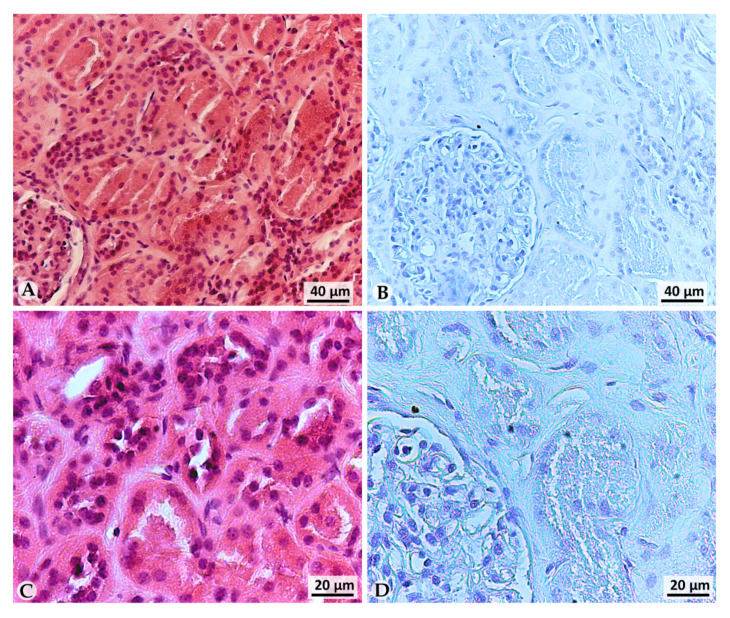
Hematoxylin/Eosin staining and LOX-1 immunohistochemistry of healthy autopsy kidney tissue. (**A**–**C**) Ematoxylin/Eosin staining, (**B**–**D**) LOX-1 immunohistochemistry, (**A**,**B**) magnification 20×, (**C**,**D**) magnification 40×.

**Figure 3 cancers-13-04213-f003:**
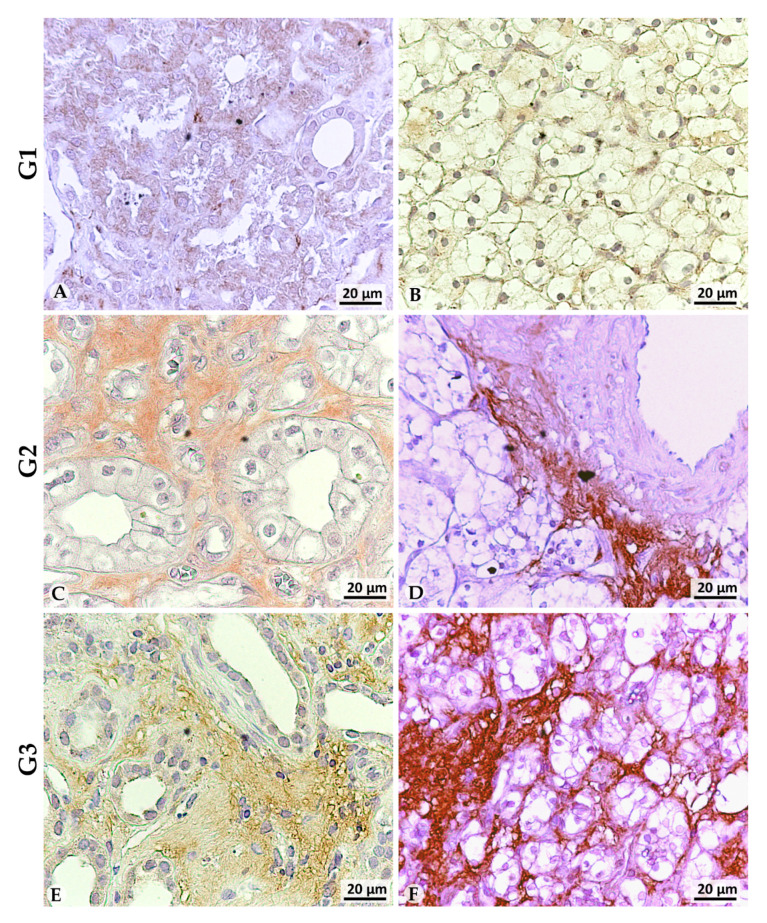
LOX-1 immunohistochemistry analysis in clear cells kidney cancer. (**A**) NpT aside G1 tumor, (**B**) G1 tumor, (**C**) NpT aside G2 tumor, (**D**) G2 tumor, (**E**) NpT aside G3 tumor, (**F**) G3 tumor. Scale bar 20 µm.

**Figure 4 cancers-13-04213-f004:**
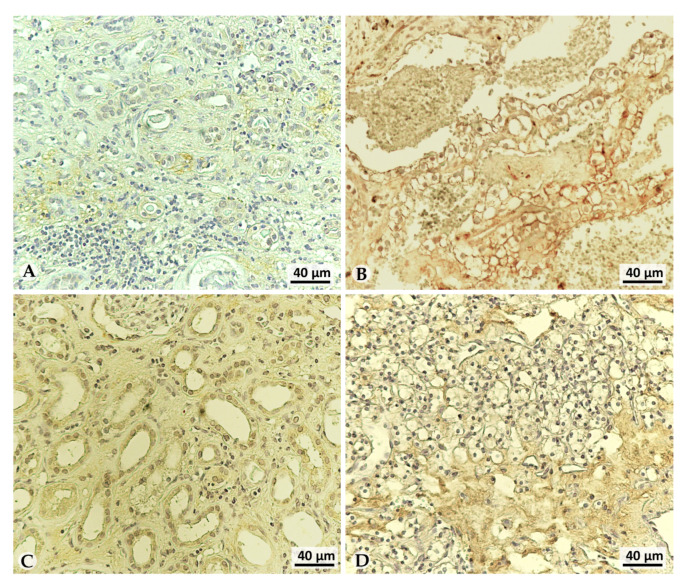
VEGF-A and LOX-1 immunohistochemistry analysis in NpT and tumoral tissues. (**A**–**C**) NpT tissues, (**B**,**D**) G2 tumoral tissue, (**A**,**B**) Immunohistochemistry of VEGF expression, (**C**,**D**) LOX-1 expression in high grade is present also in the extracellular space. Scale bar 40 µm.

**Figure 5 cancers-13-04213-f005:**
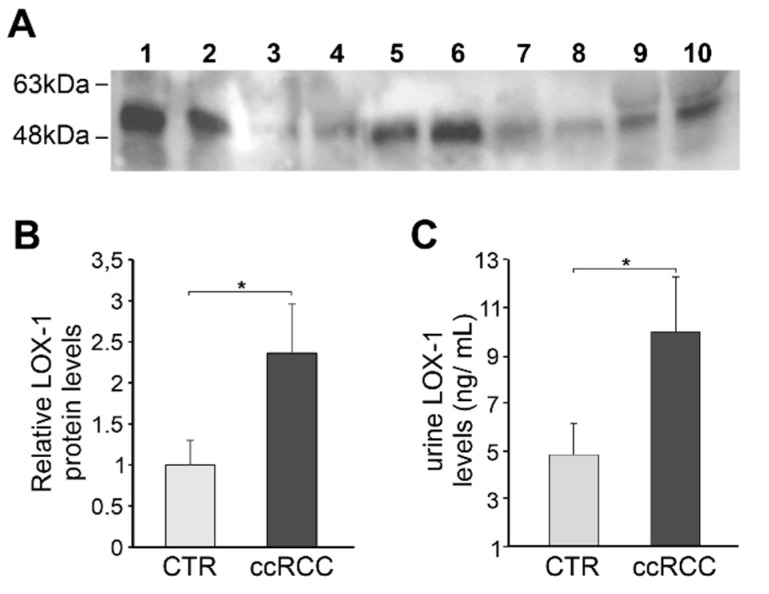
LOX-1 is revealed in urine of ccRC patients. (**A**) Representative Western blot analysis of equal amount of urine normalized with total protein from ccRCC patients (*n* = 6; line:1-2-5-6-9-10) and controls (*n* = 4; line:3-4-7-8). (**B**) Data are reported as mean ±SEM (*p*-value = 0.05) LOX-1 levels analyzed by ELISA in ccRCC urine samples (*n* = 30) and healthy controls (*n* = 25). (**C**) Data are reported as mean ± SEM (* *p*-value < 0.05).

**Figure 6 cancers-13-04213-f006:**
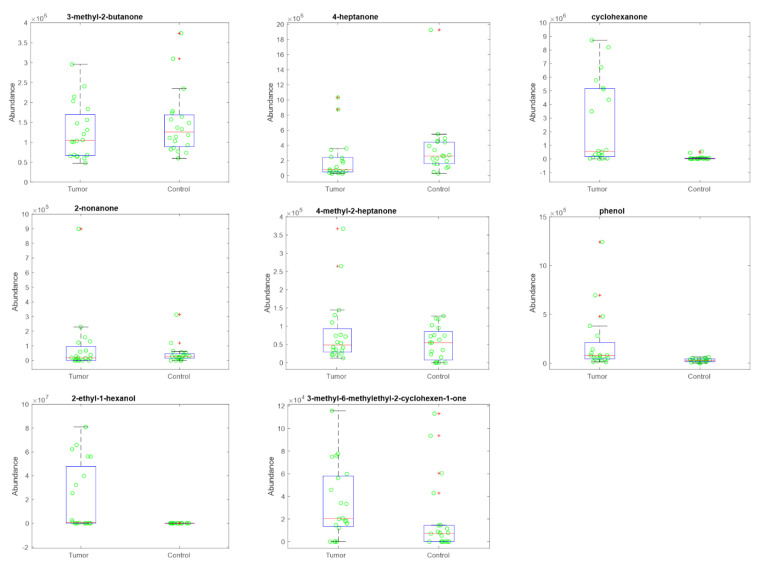
Box plots of the abundances of the most recurrent volatile compounds respect to cancer and control groups.

**Figure 7 cancers-13-04213-f007:**
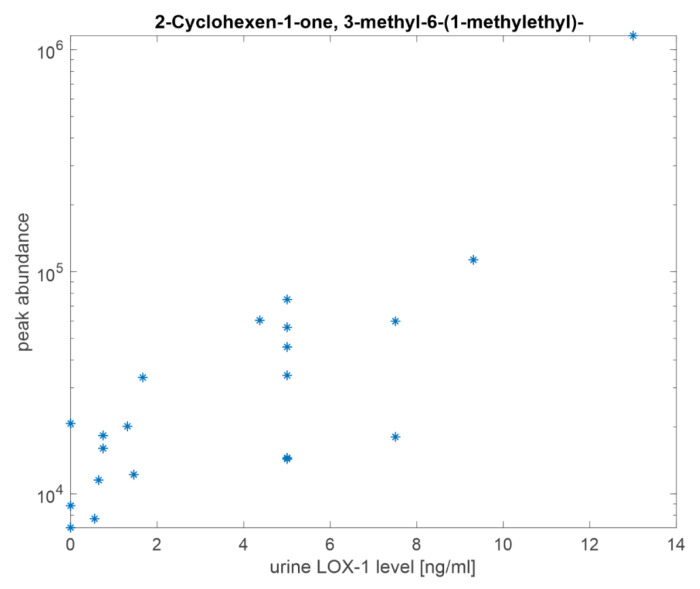
The logarithm of the abundance of 2-Cyclohexen-1-one, 3-methyl-6-(1-methylethyl)- respect the concentration of LOX-1 in urines.

**Figure 8 cancers-13-04213-f008:**
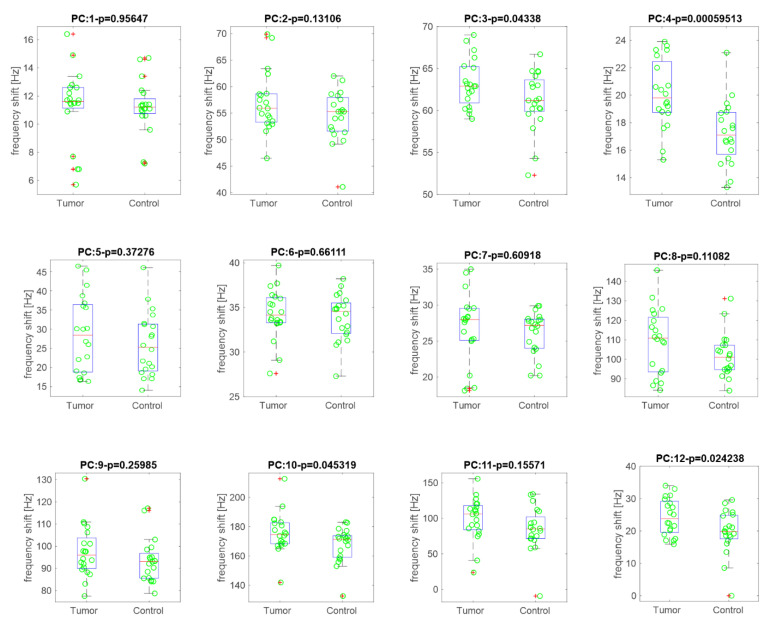
Box plots of sensor’s responses respect to cancer and control groups.

**Figure 9 cancers-13-04213-f009:**
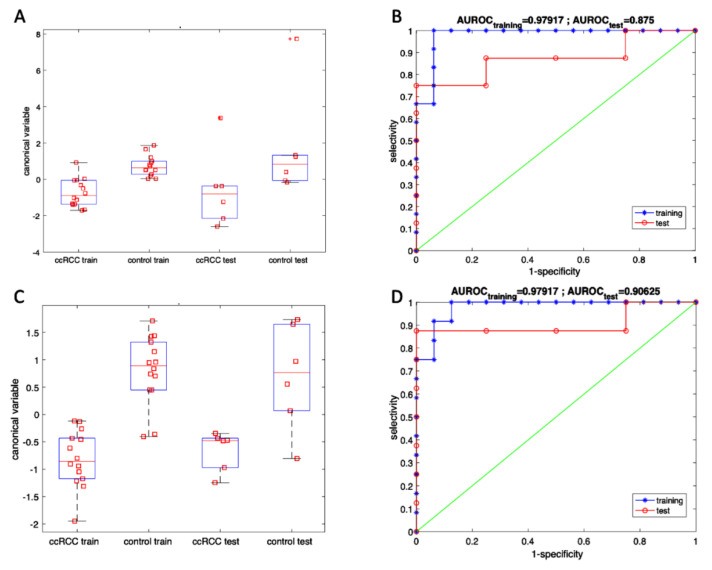
Box-plots of canonical variable (**A**,**C**) and ROC (**B**,**D**) calculated with GC/MS and sensor array data respect to cancer and control groups. Canonical variables were calculated with training data and applied to test data.

**Table 1 cancers-13-04213-t001:** Clinical-pathological characteristics.

Clinical Features	ccRCC Patients = 40
Age	62.3 yrs. (range 26–81)
Gender	
Male	24 (60%)
Female	16 (40%)
Surgery	
Partial nephrectomy	20 (50%)
Radical nephrectomy	20 (50%)
Tumor Site	
Left kidney	19 (47.5%)
Right kidney	21 (52.5%)
Tumor size	4.55 cm (range 1–11)
TNM Tumor Staging	
T1a	18 (45%)
T1b	11 (27.5%)
T2a	4 (10%)
T2b	1 (2.5%)
T3a	6 (15%)
N0-Nx	40 (100%)
N1	
M0	39 (97.5%)
M1	1 (2.5%)
ISUP Tumor Grading	
G1	8 (20%)
G2	24 (60%)
G3	8 (20%)
Tumor Necrosis	
Presence	5 (12.5%)
Absence	35 (87.5%)
Comorbidities	
Hypertension	28/40 (70%)
Diabetes	6/40 (15%)
Dyslipidemia	17/40 (42.5%)
Smoking habit	10/40 (25%)
BMI	
Weight	82.2 kg (range 56–134)
Height	170 cm (range 160–184)

**Table 2 cancers-13-04213-t002:** Distribution of LOX-1 in tumor and healthy counterpart of different grade and stage.

Patients	Grade	Normal Peritumoral Tissues (NpT)		Tumoral Tissues		
		Nucleus	Cytoplasm	Nucleus	Cytoplasm	Extracellular Space
8	G1	−	++	+	+/−	50% ++
23	G2	−	+	10% cells +/−	−	++
9	G3	−	+	−	−	++

+/− sign means weakly positive.

**Table 3 cancers-13-04213-t003:** Compounds present in at least 70% of samples.

VOC	*p*-Value
2-Butanone, 3-methyl-	0.26
4-Heptanone	0.01
Cyclohexanone	<0.001
2-Nonanone	0.79
2-Heptanone, 4-methyl-	0.55
Phenol	<0.001
1-Hexanol, 2-ethyl-	<0.001
2-Cyclohexen-1-one, 3-methyl-6-(1-methylethyl)-	0.02

**Table 4 cancers-13-04213-t004:** Distribution of LOX-1 in tumor and healthy counterpart of different grade and stage.

Analytical Instrument	Accuracy		Sensitivity		Selectivity	
	Training	Test	Training	Test	Training	Test
GC/MS	92.9%	91.7%	85.7%	83.3%	100%	100%
Sensor array	92.9%	91.7%	100%	100%	85.7%	83.3%

## Data Availability

The data presented in this study are available on request from the corresponding author.

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
