# Peer review of "Urine LOX-1 and Volatilome as Promising Tools towards the Early Detection of Renal Cancer"

_cancers, 2021, doi:10.3390/cancers13164213_

Round 1

Reviewer 1 Report

The manuscript shows a study of LOX-1 expression in a short series of CCRCC (n=40). Although short, the series is robust and the approach appropriate. Results are of potential interest to be applied in the clinical practice. For these reasond, the paper should be accepted: I found some minor comments:

  1. The Introduction section is too large, and much of the information given does not focus on the topic. I would suggest to shorten it avoiding unnecessary data non-related specifically with the objective.
  2. Remove type I and II when speaking about papillary renal cell carcinoma, since this subclassification is no longer accepted
  3. Keep terminology more correct when mentioning tumor subtypes in the introduction: papillary renal cell carcinoma and chromophobe renbal cell carcinoma.
  4. Spell correctly Hematoxylin and Immunohistochemistry throughout the figure legends

Reviewer 2 Report

The manuscript provides important information for biomarker researchers. I suggest to address the following  comments in a revised version of the manuscript.

1. Please include a Table providing the clinic pathological parameters of the study cohort.

2. Pleas add a Section on statistical analysis in the Methods section.

3. Please perform a statistical analysis of LOX-1 expression in renal tissues.

4. Did the author study corresponding volatile compounds, urine and tissue samples? Did they observe a correlation of LOX-1 expression levels in the different samples?

5. Please add ROC analyses for tissue/urine studies.

6. For validation purposes, I suggest to include an analysis of the TCGA ccRCC dataset.

Round 2

Reviewer 2 Report

The reviewer comments have been addressed, however, I recommend including the findings from the TCGA analysis into the manuscript before the manuscript should be accepted for publication

Author Response

Thank you for your suggestion.

As requested, the findings from the TCGA analysis was included into the manuscript.

best regards